# Overexpression of Canonical Prefoldin Associates with the Risk of Mortality and Metastasis in Non-Small Cell Lung Cancer

**DOI:** 10.3390/cancers12041052

**Published:** 2020-04-24

**Authors:** Xenia Peñate, Juan Manuel Praena-Fernández, Pedro Romero Pareja, María del Valle Enguix-Riego, Laura Payán-Bravo, Begoña Vieites, Lourdes Gomez-Izquierdo, Javier Jaen Olasolo, Eleonor Rivin del Campo, Jose Carlos Reyes, Sebastián Chávez, Jose Luis Lopez Guerra

**Affiliations:** 1Instituto de Biomedicina de Sevilla, Universidad de Sevilla-CSIC-Hospital Universitario V. del Rocío, Avda. Manuel Siurot s/n, 41013 Seville, Spain; xenia@us.es (X.P.); laupaybra@gmail.com (L.P.-B.); 2Department of Nursing, Universidad de Sevilla, 41009 Seville, Spain; jmpraenaf@gmail.com; 3Department of Radiation Oncology, University Hospital Virgen del Rocío, Avda. Manuel Siurot s/n, 41013 Seville, Spain; pedro.romero.pareja@gmail.com (P.R.P.); enguixriego@gmail.com (M.d.V.E.-R.); 4Department of Pathology, University Hospital Virgen del Rocío, Avda. Manuel Siurot s/n, 41013 Seville, Spain; b.vieites.pq@gmail.com (B.V.); gomezizquierdo.l@gmail.com (L.G.-I.); 5Department of Radiation Oncology, University Hospital Puerta del Mar, Av. Ana de Viya, 21, 11009 Cadiz, Spain; jjaen2008@gmail.com; 6Department of Radiation Oncology, Tenon University Hospital, Sorbonne University, 4 Rue de la Chine, 75020 Paris, France; eleonorrivin@gmail.com; 7Andalusian Center of Molecular Biology and Regenerative Medicine-CABIMER, Junta de Andalucia-University of Pablo de Olavide-University of Seville-CSIC, 41092 Seville, Spain; jose.reyes@cabimer.es

**Keywords:** prefoldin, metastasis, survival, non-small cell lung cancer

## Abstract

Canonical prefoldin is a protein cochaperone composed of six different subunits (PFDN1 to 6). PFDN1 overexpression promotes epithelial–mesenchymal transition (EMT) and increases the growth of xenograft lung cancer (LC) cell lines. We investigated whether this putative involvement of canonical PFDN in LC translates into the clinic. First, the mRNA expression of 518 non-small cell LC (NSCLC) cases from The Cancer Genome Atlas (TCGA) database was evaluated. Patients with *PFDN1* overexpression had lower overall survival (OS; 45 vs. 86 months; *p* = 0.034). We then assessed the impact of PFDN expression on outcome in 58 NSCLC patients with available tumor tissue samples. PFDN1, 3, and 5 overexpression were found in 38% (*n* = 22), 53% (*n* = 31), and 41% (*n* = 24) of tumor samples. PFDN1, 3, and 5 overexpression were significantly associated with lower OS, lower disease-free survival (DFS), and lower distant metastasis-free survival (DMFS) for PFDN1 and 3 with a trend for PFDN5. In multivariate analysis, PFDN5 retained significance for OS (hazard ratio (HR) 2.56; *p* = 0.007) and PFDN1 for DFS (HR 2.53; *p* = 0.010) and marginally for DMFS (HR 2.32; *p* = 0.053). Our results indicate that protein response markers, such as PFDN1, 3, and 5, may complement mRNA signatures and be useful for determining the most appropriate therapy for NSCLC patients.

## 1. Introduction

The high incidence of metastasis is one of the main components of mortality in non-small cell lung cancer (NSCLC). Cancer epithelial cells need to transform into mesenchymal cells to escape the primary tumor and circulate towards the target tissues where secondary tumors are established [1,2]. Epithelial–mesenchymal transition (EMT) results in the loss of epithelial markers and the increase in mesenchymal markers. During EMT, cells acquire partial pluripotency and stemness [3], particularly during intermediate states [4]. 

EMT is driven by immune cells and inflammation in the microenvironment of lung tumors [5], which activate several signaling pathways with a predominant role of the transforming growth factor-ß (TGF-ß) family of cytokines [6]. EMT in cancer is also associated with the repression of genes that promote cell cycling [7]. This repression seems to be promoted by the accumulation of prefoldin 1 (PFDN1), in response to TGF-β1-mediated activation [8]. PFDN1 is a subunit of prefoldin, a cochaperone that contributes to protein folding in cooperation with type II adenosine triphosphate-dependent chaperonins [9]. Canonical prefoldin shows a jellyfish-like structure and is composed of six different subunits (PFDN1 to 6). PFDNs also participate in processes that take place in the cell nucleus [10], including molecular events that contribute to gene regulation [11]. 

The involvement of PFDN1 in promoting EMT in lung cancer suggests that PFDN accumulation may play an important role in the evolution of lung cancer and the onset of metastasis. However, the available evidence only comes from in vitro work with cell lines and mice xenografts [8]. We report the results of a translational study that show the potential utility of PFDN in predicting overall survival (OS) and metastasis. 

## 2. Methods and Materials

### 2.1. mRNA Expression

The mRNA expression (RNA-seq) and clinical data of 518 NSCLC cases from The Cancer Genome Atlas (TCGA) database (http://cancergenome.nih.gov/) were evaluated. First, the survival plots of patients with tumors that expressed low vs. high levels of *PFDN* were analyzed. We then compared the levels of *PFDN* gene transcripts in normal tissue (*n* = 59) and the tumor tissue (*n* = 517). 

### 2.2. Patient Population

Clinical data from 58 NSCLC patients (Table 1) were collected prospectively between 2001 and 2017 and were used for this analysis. There were no notable differences in the type of chemotherapy and radiotherapy used. First-line systemic therapy consisted of platinum-based double-agent chemotherapy in all cases except one who received a protein kinase inhibitor (afatinib). A three-dimensional conformal radiotherapy technique was used in all cases except one who was treated with a volumetric arc therapy technique. Only 3 patients had epidermal growth factor receptor (EGFR) mutations, and there were not any anaplastic lymphoma kinase (ALK) translocations. In addition, only 3 patients were programmed cell death-ligand 1 (PD-L1) positive. Exclusion criteria included having either small cell lung cancer histology or previous oncologic treatments. 

### 2.3. Tissue Microarrays Immunohistochemical Analysis 

Immunohistochemical studies were performed on lung cancer specimens in a tissue microarray (TMA). The tumor samples were obtained from our Institutional Biobank and were stored in paraffin blocks of lung carcinoma. Two independent pathologists, blinded to patient data, performed tissue sampling and scored PFDN expression. Antibodies were obtained from commercially available sources: Anti-PFDN1 (ab151708) and Anti-PFDN3 (ab96085) antibodies (Abcam, Cambridge, UK); Anti-PFDN5 (sc-27119) (Santa Cruz Biotechnology, Dallas, TX, USA). The following discrete values were assigned for observations: 0, no expression; 1 (+), weak expression; 2 (++), strong expression, and 3 (+++), very strong expression (Appendix A).

### 2.4. Statistical Analysis 

SPSS (version 26.0, IBM Corp., Armonk, NY, USA) statistical software and GraphPad Prism version 5.0 (GraphPad, San Diego, CA, USA) were used for data analyses. The primary outcome was OS. The Kaplan–Meier method provided estimates of the following endpoints: OS, disease-free survival [DFS; defined as any disease recurrence (loco-regional, or distant)], loco-regional recurrence (LR), and distant metastases (DM). Multivariate analyses, including the statistically significant features in the univariate analysis, were performed using Cox’s proportional hazard model. *p* ≤ 0.05 was considered significant. 

## 3. Results

### 3.1. PFDN1 mRNA Levels Associates with OS in NSCLC 

We first analyzed the TCGA lung cohort and found that patients with high-*PFDN1* tumors had a median survival of 45 months, while patients with low-PFDN1 tumors presented a median survival of 86 months (log-rank test *p*-value = 0.034; Figure 1A). No significant differences in OS of patients with low-*PFDN* versus high-*PFDN* were found for the other five *PFDN* genes (Figure 1A). 

We also found that levels of *PFDN1-6* transcripts were significantly higher in tumors with high *PFDNs* levels than in normal tissue (Figure 1B). There were no significant differences in EGFR mutations among patients with high and low levels of *PFDN1* (*p* = 0.896), *2* (*p* = 0.549), *3* (*p* = 0.938), *4* (*p* = 0.735), *5* (*p* = 0.618), and 6 (*p* = 0.735) mRNA. ALK translocations were significantly lower only in patients with higher levels of *PFDN4* mRNA (39% vs. 61%; *p* = 0.034).

There was a high correlation between *PFDN2* and *4* mRNA levels in tumor samples (Spearman coefficient: 0.580; *p* < 0.0001) and between *PFDN2 and 6* (Spearman coefficient: 0.640; *p* < 0.0001; Figure 1C). In contrast, levels of *PFDN1* and *PFDN5* transcripts, which also correlated with each other (Spearman coefficient: 0.53; *p* < 0.0001; Figure 1C), were either invariant or increased in normal tissue with respect to tumors (Figure 1B). This discrepancy prompted us to analyze PFDN protein levels in tumor samples by immunohistochemistry. 

### 3.2. Association of PFDN and Survival in NSCLC Patients

The clinical and pathological characteristics of the patients enrolled in the study are summarized in Table 1. Median follow-up time for all patients was 40 months (range, 5 to 168 months). At the time of analysis, 67% (*n* = 39) were alive, 33% (*n* = 19) had died, and 58% (*n* = 34) had experienced a recurrence (24 were distant metastasis). 

PFDN1, 3, and 5 overexpression (+++) was found in 38% (*n* = 22), 53% (*n* = 31), and 41% (*n* = 24) of tumor samples. There was a high correlation among PFDNs levels (*p* < 0.010; Figure 2A). There were no significant differences in the stage distribution among patients with and without PFDN1, 3, and 5 overexpression (p 0.098, *p* = 0.724, and *p* = 0.796, respectively). There were 4 (18%) and 18 (82%) patients with early (I/II) and advanced (III/IV) stages, respectively, in the PFDN1 overexpressed cohort, and 14 (39%) and 22 (61%) patients with early and advanced stages, respectively, in the PFDN1 cohort without overexpression. There were 9 (29%) and 22 (71%) patients with early and advanced stages, respectively, in the PFDN3 overexpressed cohort, and 9 (33%) and 18 (67%) patients with early and advanced stages, respectively, in the PFDN3 cohort without overexpression. There were 7 (29%) and 17 (71%) patients with early and advanced stages, respectively, in the PFDN5 overexpressed cohort, and 11 (32%) and 23 (68%) patients with early and advanced stages, respectively, in the PFDN5 cohort without overexpression. Associations between corresponding patient’s clinic-pathological features and OS, DFS, local and distant relapse are shown in Appendix A. 

PFDN1, 3, and 5 overexpression were associated with lower OS (HR: 2.86, CI 1.47–5.56, *p* = 0.002; HR: 2.30, CI 1.17–4.50, *p* = 0.015; and HR: 2.94, CI 1.50–5.76, *p* = 0.002, respectively; Figure 2B–D), lower DFS (HR: 2.53, CI 1.24–5.15, *p* = 0.010; HR: 2.09, CI 1.02-4.27, *p* = 0.042; and HR: 2.002, CI 0.986–4.064, *p* = 0.055, respectively; Appendix A), and lower distant metastasis-free survival (DMFS) for PFDN1 and 3 with a trend for PFDN5 (HR: 2.94, CI 1.28–6.73, *p* = 0.011; HR: 2.50, CI 1.06–5.88, *p* = 0.036; and HR: 1.94, CI 0.85–4.42, *p* = 0.112, respectively; Figure 2E–G). The majority of patients (*n* = 48; 83%) were treated between 2012 and 2017. Similar results were found for this more recent cohort. Patients with PFDN1, 3, and 5 overexpression showed significantly lower overall survival (*p* = 0.011, *p* = 0.046, and *p* = 0.013, respectively). In addition, those with PFDN1 and 3 overexpression had also lower disease-free survival (*p* = 0.023 and *p* = 0.042, respectively) and distant metastasis-free survival (*p* = 0.013 and *p* = 0.046, respectively). Additionally, there were no significant differences (*p* > 0.05) in the type of surgery as well as in the use of radiotherapy or chemotherapy among patients with and without PFDN1, 3, and 5 overexpression.

There was no association with local recurrence. In multivariate analysis (Appendix A), PFDN5 retained significance for OS (HR 3.13; CI 1.59–6.15, *p* = 0.001) and PFDN1 for DFS (HR 2.53; CI 1.24–5.15, *p* = 0.010) and DMFS (HR 2.32; CI 0.98–5.45, *p* = 0.053; Appendix A).

When combining overexpressed PFDNs, Kaplan–Meier curves for mortality estimated OS at 24 months for 50% of patients with PFDN1, 3, and 5 overexpression (HR: 2.550; CI: 1.280–5.090; *p* = 0.008; Figure 2H) and 86% for the rest of patients. Two-year DFS rate for patients with PFDN1 and 3 overexpression was 35% vs. 70% for those with lower expression (HR: 2.370; CI: 1.140–4.900; *p* = 0.020; Appendix A). In addition, patients with PFDN1 and 3 overexpression had a 2-year DMFS rate of 49% vs. 78% in the subset of patients with lower expression (HR: 2.880; CI: 1.250–6.600; *p* = 0.012: Appendix A). 

## 4. Discussion

Overexpression of canonical PFDN associates with the risk of mortality and metastasis in non-small cell lung cancer (LC). Our pertinent findings indicate that patients with *PFDN5* overexpression in the tumor tissue had higher mortality rates, and those having *PFDN1* overexpression also showed higher rates of recurrence, specifically for distant metastasis. When combining overexpressed PFDNs, patients with *PFDN1*, *3*, and *5* overexpression had lower OS, and those with *PFDN1* and *3* overexpression had lower DFS and DMFS rates. These results are in agreement with the high correlation among PFDN subunits in the tumor tissue. 

PFDN2 and 6 are also present in the unconventional prefoldin RPB5 interactor (URI)-prefoldin-like complex, which has been related to several types of cancer but not LC [12]. However, PFDN1, 3, and 5 are not present in the Uri-complex, showing that the detected clinical effects in LC are mediated by canonical PFDN, likely mediated by its nuclear role acting on cyclin A expression, as proposed by Wang et al. [8]. PFDN1 suppressed cyclin A expression by directly interacting with the cyclin A promoter at the transcriptional start site. Strikingly, cyclin A overexpression abolished the above PFDN1-mediated effects on the behavior of lung cancer cells, whereas cyclin A knockdown alone induced EMT and increased cell migration and invasion ability [8]. We conclude that overexpression of canonical PFDN is a prognostic marker of mortality and metastasis in NSCLC.

We acknowledge several limitations of our study. First, the small number of patients enrolled in the study was heterogeneous, having been treated over more than a decade, during which time advances in imaging and therapeutic regimens have occurred. Additionally, patients received different combinations of surgery ± chemoradiation therapy rather than being treated according to a well-defined treatment protocol. To reduce the risk of bias, we analyzed the effect of PFDN overexpression in a more recent cohort and found similar results. Moreover, the type of surgery, radiotherapy, or chemotherapy among patients with and without PFDN1, 3, and 5 overexpression was assessed, and there were no significant differences. Currently, our group is expanding this study in a larger and more homogeneous cohort of patients recently treated. The influence of common lung cancer mutations (i.e., EGFR, ALK, etc.) or programmed cell death-ligand 1 (PD-L1) in the PFDN expression could be elucidated in this further study.

Poor genome-wide correlation between expression levels of mRNA and protein has been reported [13,14]. Lack of correlation between *PFDN* mRNA and PFDN protein levels, due to translational or protein turn-over regulation, may explain why PFDN was not included in any of the expression signatures so far proposed for the prognosis of LC based on transcriptomics (see Biswas et al. [15] and the other nine previous signatures cited therein). Other EMT protein markers with high prognostic significance in NSCLC, such as Snail and TWIST1 [16], were also out of published prognostic transcriptomic signatures. Our results indicate that protein response markers, such as PFDN1, 3, and 5, may complement mRNA signatures and be useful for determining the most appropriate therapy for NSCLC patients.

## 5. Conclusions

Our results show that overexpression of canonical PFDN associates with the risk of mortality and higher rates of recurrence, specifically for distant metastasis, in NSCLC. This response marker may complement mRNA signatures and be useful for guiding intensity in individualized therapy for NSCLC patients.

## Figures and Tables

**Figure 1 cancers-12-01052-f001:**
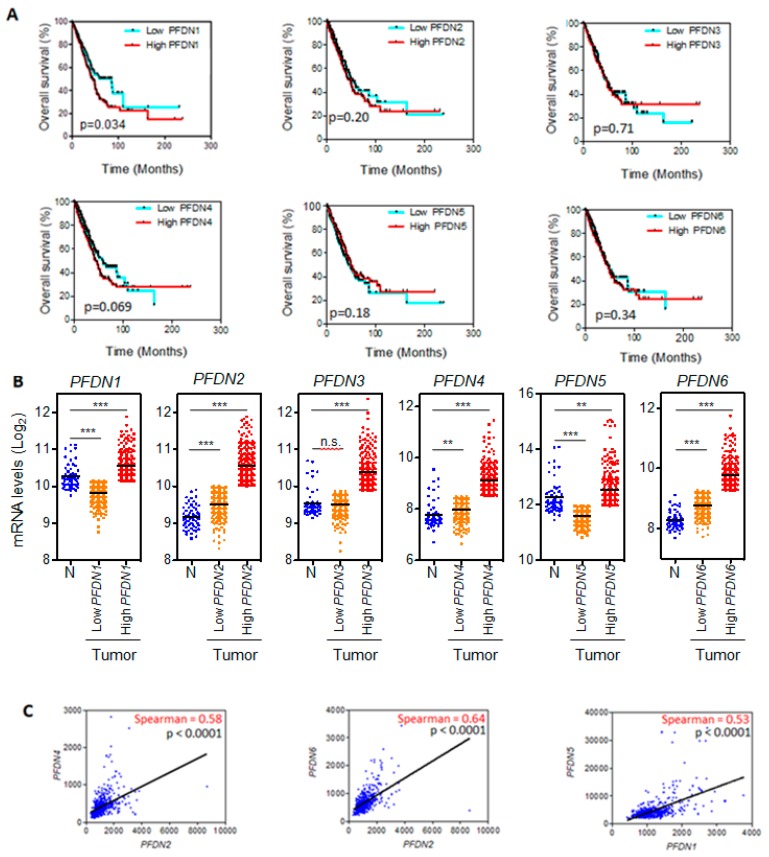
(**A**) Kaplan–Meier curve of The Cancer Genome Atlas (TCGA) lung cohort for overall survival according to low vs. high expression of *prefoldins* (*PFDNs)* (cut off point: 50%); (**B**) Levels of *PFDNs t*ranscripts in tumors and normal tissue (Abbreviations: N, normal; n.s., not significant; *** *p* < 0.0001; ** *p* < 0.001). Tumor samples were divided into two categories: low and high expression of *PFDNs* as in (**A**); (**C**) Significant correlations between mRNA levels of different *PFDN* genes in tumor samples. RNA-seq expression data are provided as RNA-Seq by Expectation Maximization (RSEM) normalized data.

**Figure 2 cancers-12-01052-f002:**
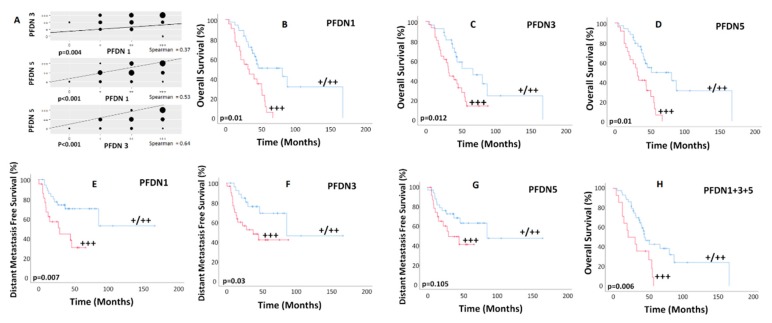
(**A**) Correlation among PFDN protein levels for all 58 patients. The diameter of the dots is proportional to the number of cases. (**B**–**H**) Kaplan–Meier curves for overall survival (**B**–**D**) and distant metastasis-free survival (**E**–**G**) according to PFDN expression and overall survival when combining all overexpressed PFDNs (**H**).

**Table 1 cancers-12-01052-t001:** Patient characteristics.

Characteristics	NO. of Patients (%) *n* = 58
Gender	
Female	8 (14)
Male	50 (86)
Age, years	
Median (range)	67 (41–82)
Chronic obstructive pulmonary disease	
No	34 (59)
Yes	24 (41)
Hypertension	
No	31 (53)
Yes	27 (57)
Diabetes Mellitus	
No	42 (58)
Yes	16 (42)
Dyslipidemia	
No	26 (45)
Yes	32 (55)
Cardiovascular disease	
No	45 (78)
Yes	13 (22)
Thrombosis	
No	49 (84)
Yes	9 (16)
Smoking status	
Never	3 (5)
Former	28 (48)
Current	27 (47)
History of alcohol consumption	
No	31 (53)
Yes	27 (47)
Karnofsky Performance Status	
100	13 (22)
90	15 (26)
80	16 (28)
70	14 (24)
Histology	
Adenocarcinoma	24 (41)
Squamous	34 (59)
T stage	
T1	13 (22)
T2	28 (48)
T3	11 (19)
T4	6 (10)
N stage	
N0	19 (33)
N1	8 (14)
N2	25 (43)
N3	6 (10)
M stage	
M0	53 (91)
M1	5 (9)
Stage	
IA	1 (2)
IB	13 (22)
IIA	4 (7)
IIIA	27 (47)
IIIB	8 (14)
IV	5 (9)
Surgery	
Yes	52 (90)
No	6 (10)
Thoracic radiation therapy *	
Yes	24 (41)
No	34 (59)
Chemotherapy **	
Yes	40 (69)
No	18 (31)

* Delivery with radical intent; ** Delivery with radical intent in all cases except one.

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
