# Peer review of "Overexpression of Canonical Prefoldin Associates with the Risk of Mortality and Metastasis in Non-Small Cell Lung Cancer"

_cancers, 2020, doi:10.3390/cancers12041052_

Round 1

Reviewer 1 Report

I suggested, for the future to include also patients with ros-1 traslocation together with ALK mutated patients. Moreover authors could analyze patients with common or uncommon EGFR mutations, more or less resistant to target therapy

Reviewer 2 Report

I reviewed the revised version of the paper and afterwards carefully evaluated the responses to the Referee I. My first opinion was that the study has several limits mainly related to the small number of patients and the different treatments that patients received. However, in the discussion the authors underlined these limitations pointing out the need to expand the study and to evaluate also more recent therapeutic strategies such as immunotherapy in the next future. On the other hands the referee who previously revised the manuscript addressed important concerns to the authors (the different clinical stages among patients, the different treatments, the need to study a more homogeneous group of patients). For the majority of them the authors have given explanations and added comments in the text. Being a brief report, and considering that the association of PFDN expression with survival parameters is a new finding, I think that in the present form the paper should be accepted for publication as brief report as indicated.

This manuscript is a resubmission of an earlier submission. The following is a list of the peer review reports and author responses from that submission.

Round 1

Reviewer 1 Report

The brief report investigates a possible association of PFDN expression and survival parameters in NSCLC patients. In a sample of 58 patients of different clinical tumor stages PDFN 1,3,5 overexpression were detected to variing extent in IHC.

The authors state that the overexpression of PDFN 1,3 and 5 was associated with a lower OS, DMFS and DFS in their patient cohort.

The major problem of this work is that the overexpression was measured in NSCLC specimens of patietns with very different clinical stages including stage IIIB and IV. Are these patients evenly distributed in the PDFN negative and positive cohorts?

The patients were treated in a time span from 2001 - 2017. Beside changes in the staging system, therapeutic regimens such as surgical techniques, radiation therapy and of course chemotherapy has been changing and improving significantly in this time frame. Can the authors exclude different therapy approaches as a possible bias?

In Table S1: How do the authors explain that there is no significant difference regarding OS and DFS between stage I/II and stage III/IV patients?

Table S1: Did the authors differentiated between palliative chemotherapy and concomitant/sequential chemotherapy?

Furthermore was there a distinctinon between palliative and curative radiotherapy?

In table 1 patients with/without thrombosis add up to 60 patients.

The question if PFDN overexpression might have an influence on survival parameters in NSCLC patients or may even bepredictive for some kind of antitumortherapy is very interesting. However it might be beneficial to investigate this question in a mor homogeneous patient collective. 

Reviewer 2 Report

Authors proved the correlation between three prefoldin subunits and poor prognosis of patiens with NSCLC.

Major comments:

Fig. 1 Authors should include an analysis of PFDN1 expression comparing normal tissue, tumors with high PFDN1 and tumors with low PFDN1. Moreover they could include Overall survival analysis also for other subunits

Authors should analyze the correlation between the expression of PFDN subunits and common lung cancer mutations (e.g. EGFR, ALK, ecc.) or PD-L1. For my experience NSCLC cells with EGFR mutations have an elongated morphology so I mean that suggested analysis could improve the relevance of this paper.

Minor comments:

Abstract: authors should introduce the concept that canonical prefoldin includes several subunits

Introduction:

line 45 change "no-small cell lung" in " non-small cell lung"

line 48 after "established" include a reference

line 51 Look the style of "EMT is driven by immune cells"

Figures

Figure 2: improve the picture because it has a bad screen resolution